# MRI Anatomical Investigation of Rabbit Bulbourethral Glands

**DOI:** 10.3390/ani13091519

**Published:** 2023-04-30

**Authors:** Rosen Dimitrov, Kamelia Stamatova-Yovcheva

**Affiliations:** Department of Veterinary Anatomy, Histology and Embryology, Faculty of Veterinary Medicine, Trakia University, 6000 Stara Zagora, Bulgaria; rdimitrov288@gmail.com

**Keywords:** imaging anatomy, male accessory sex glands, animals

## Abstract

**Simple Summary:**

The transverse Magnetic Resonance Imaging (MRI) of rabbit bulbourethral glands has achieved detailed anatomical information. The quality of the anatomical tissue contrast has so far been similar in both sequences in the dorsal MRI of bulbourethral glands. The sagittal MRI of the glands has demonstrated a variety of anatomical contrast independent of the image plane. Rabbit accessory sex glands have been evaluated as suitable biological objects to demonstrate the effect of the application of imaging and diagnostic protocols to investigate the reproductive morphology and pathology of animals. The obtained imaging information by MRI could be used for the differentiation of the studied glands from adjacent soft tissue structures in the retroperitoneal part of the pelvic cavity. Recently MRI has been involved as a relatively new and dynamically developing imaging method for the diagnostics of the urogenital system in different animal species. It is essential to obtain detailed anatomic data that are necessary for the interpretation of imaging findings.

**Abstract:**

Anatomical MRI is appropriate for the interpretation of soft tissue findings in the retroperitoneal part of the pelvic cavity. The aim of the current study was to use rabbits as an imaging model to optimize MRI protocols for the investigation of bulbourethral glands. The research was conducted on twelve clinically healthy, sexually mature male rabbits, eight months of age (New Zealand White), weighing 2.8 kg to 3.2 kg. Tunnel MRI equipment was used. The transverse MRI in the T2-weighted sequence obtained detailed images that were of higher anatomical contrast than those in T1-weighted sequences. The hyperintensity of the glandular findings at T2, compared to the adjacent soft tissues, was due to the content of secretory fluids. The quality of the anatomical tissue contrast has not shown much dependence on the choice of the sequence in dorsal MRI. The sagittal visualization of the rabbit bulbourethral glands corresponded to the localization of the research plane toward a median plane. The imaging results could be used as a morphological base for clinical practice and reproduction.

## 1. Introduction

Cowper’s glands have been found in almost all placental animals. Their absence occurs in dogs, insectivores and cetaceans. In marsupials, three parts of the glands have been observed, while in monotremes, only accessory glands have been observed [1,2].

The bulbourethral glands (*glandulae bulbourethrales*, Cowper’s glands) are paired accessory sex glands that protect the urethral epithelium. Their secretion contributes to the coagulation of seminal fluids. They are yellow-brown in color and have a flattened bean shape, as the right gland is relatively larger than the left gland. The glands are located in the bulbospongiosus muscle at the caudal end of the pelvic part of the male urethra. They are included in the pelvic diaphragm. Their main excretory ducts drain into the bulbar part of the urethra, passing through its spongy layer [1,2,3,4,5,6,7]. Rabbit accessory sex glands have been used as suitable biological objects to demonstrate the efficiency of applying imaging and diagnostic protocols in reproductive morphology and pathology. Rabbit bulbourethral glands have been applied as a topographic landmark for the division of the pelvic and penile urethra [7]. Rabbit bulbourethral glands have been visualized anatomically (via computed tomography, CT) as cubic, solid, soft tissue findings. They have been found between the dorsal urethral wall and the ventral rectal wall and connected to the paraprostatic parts by soft tissue [8,9]. The thickness of the obtained CT anatomical sections has been 2 mm, and the glands have been involved in transversely oval homogeneous findings with soft tissue X-ray attenuation. They are distinguishable from the peripheral soft tissues of the retroperitoneal pelvic space. A CT image has been found in the transverse plane through the cranial part of the second caudal vertebra (dorsally), the body of the ischium (laterally) and the sciatic arch (ventrally) [8,9].

Magnetic Resonance Imaging (MRI) of the accessory sex glands has been evaluated as a relatively new and dynamically developing imaging method for urogenital morphology and urology [10]. The rabbit has been chosen as an animal model for the compilation of an MRI atlas using transverse, dorsal and sagittal T1- and T2-weighted images [11]. Using this method, the rabbit prostate complex has been visualized as a hyper-intense finding on T1-weighted sequences. The capsular part has been hyper-intense compared to the glandular part. A glands image has been found between the transverse planes from the caudal part of the first sacral vertebra to the cranial part of the third sacral vertebra. The hyperintensity of the prostate complex has been higher on T2-weighted images. The relative hyperintensity of the prostate part, compared to the hypointensity of the proprostate and paraprostate parts, has demonstrated the important role of the prostate part in the secretory function of the glandular complex [12]. An MRI anatomical study of the urethra and periurethral tissues has been used in the morphological investigation of urethral anomalies in humans, such as Syringocele, Cobb’s collar, cowperitis, lithiasis and cancer [13,14,15,16,17,18,19,20]. In an MRI (T2-weighted images) of the urogenital diaphragm in the man, the bulbourethral glands appeared as encapsulated, heterogeneous findings with intermediate intensity. The detection of this intensity has been necessary to differentiate the glands from prostate hypertrophy or cancer [21].

MRI has been essential in obtaining the anatomic results, from which it is important to interpret a number of pathologic imaging findings. The major use of MRI protocols and devices in small animals has been due to their relative affordability and lower operating costs. Simultaneously the risk has been lower in accordance with the application of new methods and constraint codes when generating pulsed imaging programs. The methodological MRI transition from the imaging of biological objects to clinical objects has been difficult. Therefore, the system algorithm could be significantly shortened by using initial programming, optimization and the testing of new method setups [22]. Compared to the other imaging methods, this non-invasive technique allows excellent soft tissue differentiation, whole-body data and functional imaging to be obtained. Thus, the MRI of rabbit bulbourethral glands has been appropriate either for the anatomical algorithm or for imaging differentiation from adjacent soft tissue structures and pathological alterations [23]. 

The aim of the current study was to use the rabbits as a model to optimize MRI protocols for the imaging of bulbourethral glands in other species for research. The obtained information could also be applied to clinical purposes.

## 2. Materials and Methods

The experimental animals used in this study were twelve clinically healthy male live rabbits aged eight months (New Zealand white breed), with a weight range from 2.8 kg to 3.2 kg. Anatomical MRI was performed in accordance with the permissions of the Animal Ethics Committee, Faculty of Veterinary Medicine, Trakia University, Stara Zagora (No. 51 of 29 September 2012 and No. 59 of 17 May 2013), the provisions of the Law on the Protection of Animals in Bulgaria (promulgated in the State Gazette, No. 13 of 2 August 2008), and European Convention for the Protection of Vertebrate Animals Used for Experimental and Other Scientific Purposes (ETS No. 123). The current study included only live healthy animals delivered from the farm of the Agricultural Institute, Stara Zagora, Bulgaria. Before the study, these animals were examined by veterinarians working on the farm. At the end of the study, the rabbits were returned to the institutional farm.

The animals were housed in an environment with a temperature of 25 °C, with a 12 h light and 12 h dark circadian cycle. Their diet was standard, and food access was restricted for six hours before the study; water access was free [12]. The rabbits were anesthetized with 15 mg/kg Zoletil^®^ 50 (Virbac, Carros, France) (IM) (tiletamine hydrochloride 125 mg and zolazepam hydrochloride 125 mg in 5 mL of the solution) (Virbac, France). Anesthesia was potentiated with a Ketaminol^®^ 10 solution (Intervet, Unterschleißheim, Germany) (IM) (Ketamine hydrochloride 100 mg/mL and Benzethonium chloride 0.1 mg/mL), in a dose of 0.5 mL/kg [12].

MRI was performed using a tunnel MR scanner (1.5 T, Magnetom Essenza, version of software—Tim+Dot, Siemens Healthcare, USA, Whole body imaging; Dot^®^, The Siemens MRI software, Ferndale, MI 48220). The animals were positioned horizontally in the supine position on a flat table, as the pelvis was positioned in the isocenter of the magnet. Immediately before the study, a spasmolytic medication (Ketonal, Lek Pharmaceuticals d.d., Verovškova 57, Ljubljana, Slovenia, 100 mg/2 mL solution for injection, 1 mg/kg i.m.) was applied to eliminate intestinal peristalsis, which could worsen image quality [24,25]. The retroperitoneal pelvic space and the bulbourethral glands were studied in transverse, dorsal and sagittal planes at a distance from the seventh lumbar vertebra (L7) to the third caudal vertebra (Cd 3). The section thickness was 2 mm [12].

The used images were collected adequately and precisely from all twelve studied animals. The most qualitative are presented.

MRI slices were aligned on the following bones and soft tissue objects: bone anatomical landmarks—on transverse sections, depending on the topography of the slices to the corresponding vertebra, parts of the ischium and the caudal part of the pelvic symphysis; on dorsal sections—the topography of the slices to the pelvic symphysis; on sagittal sections—the topography of the slices to the spine and to the median plane. Soft tissue anatomical landmarks: on transverse sections—rectum and penile bulb (dorsal); pelvic part of the urethra (ventral); prostate complex (cranial); pelvic diaphragm; and penile root (caudal).

The study was performed under the following research imaging protocol: a magnetic field strength of 1.5 T; a superconducting type magnet; the diameter of the magnetic cylinder at 70 cm; 4—Channel Special-Purpose coil elements; matrix 256 × 256; pixel 1 mm^2^; transversal, sagittal and dorsal anatomical images—weighted in T1 and T2 spin echo sequences; 2D acquisition schemes were applied for the sequences; FOV was 50 cm^3^ (with mean values 250/250), in all directions; SNR was 20 dB; echo time (TE) was 14 ms for T1 and 90 ms for T2; the repetition time (TR) was 500 ms for T1 and 4000 ms for T2; the voxel size was 10 mm^3^; and the pelvis was scanned with a full urinary bladder [12,26]. The size of the voxel was increased to identify the number of tissue components of the glands. At the same time, the number of nuclei increased parallel to the SNR [25].

## 3. Results

The imaging results included only qualitative information, which is necessary for the imaging anatomical identification of rabbit bulbourethral glands. The anatomical relationships of the rabbit bulbourethral glands were determined for the different planes—transverse, dorsal and sagittal. They were interpreted in accordance with bone tissue and soft tissue anatomical landmarks.

### 3.1. Subsection Transverse MRI Visualization

On transverse T2-weighted imaging in the corresponding planes (first caudal vertebra, then the plane between the first and second caudal vertebrae—dorsal; the caudal part of the pelvic symphysis—ventral; the plate of ischium—lateral), the rabbit bulbourethral glands were identified. They appeared as soft-tissue, homogeneous organs with hyper-intense characteristics compared to the peripheral soft tissue landmarks (urethra, rectum, pelvic diaphragm), except for the penile bulb and the beginning of the penile root. The latter were visualized as findings with the highest intensity relative to structures in the perineum. The shape of the glands was oval. The glandular capsule was differentiated as a hypo-intense peripheral ring-like finding. The ventrally located pelvic part of the urethra was hypo-intense and imaged dorsally to the caudal part of the pelvic symphysis. The rectum showed a hypo-intense image that was localized dorsally to the penile bulb and root and ventrally to the first caudal vertebra (Figure 1).

On transverse MRI in a T1-weighted sequence of the pelvis, planes through the following bone anatomical landmarks were used: the first and second caudal vertebrae, dorsally, the caudal part of the pelvic symphysis, ventrally, and the plate of the ischium, laterally. The bulbourethral glands were difficult to identify. They were visualized as soft-tissue, homogeneous findings with an intermediate characteristic compared to the peripheral organs (urethra, rectum, pelvic diaphragm). The shape of the glandular findings was not clearly defined. Each of the glands was imaged independently of the adjacent structures but with indistinct organ boundaries, capsules and parenchyma. The ventromedial pelvic part of the urethra was hypo-intense and visualized dorsally to the caudal part of the pelvic symphysis but without any clear definition of the studied glandular structures. The urethral lumen was hypo-intense relative to the urethral wall. The lumen of the rectum was the most hypo-intense finding compared to the rest of the organs in the perineum (Figure 2 and Figure 3).

### 3.2. Subsection Dorsal MRI Visualization

Dorsal imaging in the pelvis (at the greatest distance, dorsal to the pelvic symphysis) on T1-weighted sequences demonstrated the bulbourethral glands as homogeneous findings with low or intermediate intensity similar to that of the urethra. The intensity of the adipose tissue was used as a relative marker for the degree of tissue intensity in the studied subjects. The hypo-intense character of the glandular findings was defined as the relatively high intensity of the prostate complex and the penile root. The shape of the studied glands was oval, but their borders (capsule) were not sufficiently defined. The glands were located lateral to the caudal part of the pelvic urethra, caudal to the prostate complex and cranial to the root of the penis (Figure 4).

On the dorsal imaging of glands in T1-weighted sequences, at a relatively middle distance dorsal to the pelvic symphysis, the glandular findings could clearly be defined by the hypo-intense part of the pelvic urethra (ventromedial) and the neck of the urinary bladder (cranial). The bulbourethral glands were located caudal to the prostate complex and dorsolateral to the pelvic part of the urethra. Their shape was oval. The intensity of the bulbourethral glands was the lowest compared to that of the prostate complex. The image of the vesicular glands had the highest intensity compared to other accessory sex glands (Figure 5).

The dorsal imaging of the pelvic findings was on T2-weighted sequences but at the closest distance to the pelvic symphysis, which demonstrated the ventral parts of the bulbourethral glands located on the lateral wall of the pelvic urethra. The latter were relatively hyperintense, homogeneous, with elongated craniocaudally soft tissue structures, compared to the vesicular glands and the prostate glandular complex. The pelvic urethra was hypo-intense compared to the ventral parts of the bulbourethral glands, prostate complex and vesicular glands. The shape of the ventral parts of the bulbourethral glands was craniocaudally elongated and oval, with poorly defined borders (capsule). The glandular finding was visualized as a compacted, thickened and hyper-intense part of the urethral wall (Figure 6).

The T2-weighted dorsal image of the bulbourethral glands at the pelvic symphysis demonstrated the ventral portions of the glandular findings to be hyper-intense and homogeneous compared to the cranially located ventrocaudal parts of the prostate complex and pelvic urethra. The glands were presented as tissue-defined, irregularly oval perineal structures located at the caudal end of the pelvic urethra and cranially from the beginning of the penis. The studied glands were found in the central part of the perineum, with a characteristic intensity that could distinguish them significantly from the close soft tissues in the perineal region (Figure 7).

### 3.3. Subsection Sagittal MRI Visualization

On the sagittal imaging of bulbourethral glands on T2-weighted sequences, the glandular findings could be visualized as soft-tissue, hyper-intense (compared to the rectum) and homogeneous structures. The studied glands showed a low intensity compared to the hyper-intense findings of the prostate complex, pelvic diaphragm and penile root. They had a higher intensity compared to the hypo-intense vesicular glands. The perineal location of the bulbourethral glands close to the pelvic diaphragm, penile root and rectum was determined. Topographically, the studied glandular findings were observed in the area of the pelvic outlet, at the caudal end of the pelvic urethra and ventrally to the bodies of the first and second caudal vertebrae. The image of the glands was oval and had a lack of differentiation between the stroma and parenchyma (Figure 8).

The sagittal imaging of the bulbourethral glands in T1-weighted sequences presented the glandular findings as soft-tissue, irregularly oval, with unclearly defined borders (capsule) and parts (stroma and parenchyma). The glands presented increased intensity compared to that of the rectum and decreased intensity compared to that of the pelvic diaphragm. The bulbourethral glands had a higher intensity than the vesicular glands and the prostate complex. The glandular image was found in the perineal region, cranial to the root of the penis, and at a close distance to the pelvic diaphragm. Topographically, the localization of the glandular findings was visualized at the outlet of the pelvic cavity, ventral to the first two caudal vertebrae and the rectum (Figure 9).

## 4. Discussion

In recent decades, CT, MRI and, to a lesser extent, Nuclear Medicine have been increasingly used [27]. MRI has provided detailed images with a better soft tissue contrast compared to CT [28]. The most referenced EU Directive (Directive 2010/63/EU on the protection of animals used for scientific purposes) has stated that animals have intrinsic values that need to be respected. Animal welfare considerations should be given the highest priority as per the directive, and each user needs to be carefully evaluated. The 3Rs (replacement, reduction, refinement) are considered systematically when using animals in biomedical research. In this aspect, MRI contributes to the reduction in animal numbers and allows for the refinement or even replacement of more invasive experimental techniques [29].

The rabbit as a research model has been recommended because of easy availability and housing management. Rabbit breeding is widely practiced as it can produce a large number of offspring in a short time with ease of care and the production of dietary meat that is of high quality. The accessory sex glands have a crucial role in the production of semen with high quality and its transport to the female copulatory organ during ejaculation. A limited number of studies about MRI investigations of the genital tract of male rabbits are available [30].

Complete, definitive, cross-sectional anatomical images of the bulbourethral glands were obtained using an MRI of the pelvis in the plane marked by the first and second caudal vertebrae (dorsally), the ischial bone (laterally) and the pelvic symphysis ventrally, the results of which were in partial agreement with a CT study on these glands in the rabbit. The MRI study has presented detailed data on the soft tissue features and spatial relationships of investigated glandular findings compared to the results of their CT imaging [8,9]. Morphological details in dorsal imaging of the bulbourethral glands have been dependent on the height of the dorsal scanning plane relative to the pelvic symphysis. Decreasing the distance between this plane (dorsally) and the pelvic symphysis (ventrally) caused a fragmentation of the glandular images with the representation of only the ventral glandular parts, which were small and poorly defined. Excessively raising the height of the dorsal plane has led to the visualization of dorsal parts of the glandular image or its absence. In the rabbit, the vesicular glands, prostate and bulbourethral glands, forming the accessory sex glands, were located dorsal to the pelvic part of the male urethra from the cranial to the caudal direction [30]. The quality of the anatomical tissue contrast has not presented much dependence on the choice of the sequence (T1 or T2) in dorsal imaging of the bulbourethral glands. The hypo intensity of the investigated glandular findings (dorsal MRI), compared to that of the prostate complex and vesicular glands, was a consequence of the significant development of the stroma and the weaker development of their parenchyma compared to the rest of the accessory sex glands. The sagittal MRI study of the bulbourethral glands in the rabbit demonstrated that the pelvic diaphragm did not encompass the glandular findings but remained caudal to them. Therefore, the present study has defined the studied glands as soft tissue structures separated from the soft tissues of the caudal part of the pelvic floor. This contradicted the localization of the glands in the pelvic diaphragm in the man [1,2,3,4,5].

An MRI anatomical study of the bulbourethral glands in the rabbit demonstrated their bilateral organ localization. At the same time, the glands’ shape was longitudinally oval. The intrapelvic caudal localization of the glands in the bulbospongiosy tissue was similar to that in the man, except that they were differentiated from the soft tissues at the pelvic outlet [1,2,3,4,5].

In the rabbit, the connective tissue glandular capsule has been identified on transverse images, but additional findings of the bulbourethral glands have not been visualized in the penile bulb region. This can be seen in contrast to the sporadically observed secretory structures in men and some animals [1,2,3,4,5]. The obtained images present morphological features that are important when differentiating the studied glands from similar glandular tissues in the retroperitoneal pelvic space. The results of the MRI study have visualized the glands as voluminous solid findings without tube characteristics. The MRI anatomical identification of the bulbourethral glands in the rabbit has been essential in detecting pathological alterations related to glandular morphology, such as the obstruction of the glandular excretory ducts (*Syringocele*) [1,2,3,4,5,6]. The obtained imaging data supported the hypothesis of many authors regarding the partial morphological correspondence of Cowper’s glands in humans and rabbits. This supports the argument for the importance of the rabbit as an animal anatomical model in reproductive morphology [7,31,32,33,34].

Rabbit bulbourethral glands are green paired longitudinally oval structures found in the cranial part of the sciatic arch, ventral to the root of the penis, and dorsal to the caudal part of the rectum [30]. It has been determined that the secretion produced by bulbourethral glands is directly drained into the spongy part of the male urethra. The pelvic localization of the rabbit bulbourethral glands has been caudally connected to the urethra and paraprostate and has provided a soft tissue marker for the definition of the pelvic urethra’s separate parts. This corresponds to the information established by anatomical methods of dissection [7,8]. Transverse MRI findings on the dorsal glandular surface have demonstrated a distinguished, hypo-intense, median groove separating the right from the left gland. These results complement previous anatomical data [7,8,34]. The MRI of bulbourethral glands in the rabbit, described as longitudinally oval capsulated findings, corresponds to the data obtained for these glands by CT [15,16]. Similar to the previous imaging studies, the present MRI confirmed the topographic features of these soft tissue findings. The results of the MRI, identifying the imaging features of the glands, according to the bone and soft tissue landmarks in the pelvic region, corresponded partially with those described by some authors [15,16] regarding the CT visualization of these organs in the rabbit.

The study visualized the localization of the glands as topographically dependent, both from the respective caudal vertebra (dorsal), caudal part of the pelvic symphysis (ventral) and ischial plate (lateral), as well as from the rectum (dorsal), penile bulb, pelvic diaphragm (caudal) and pelvic urethra (ventral). Therefore, the imaging results correspond with complemented and detailed results [15,16] on the CT anatomy of rabbit bulbourethral glands.

The algorithm for the MRI anatomical study of rabbit bulbourethral glands corresponds to the MRI of the prostate complex in the rabbit, regarding the application of an appropriate imaging protocol when determining the localization, topography, shape and soft tissue characteristics of the objected findings [17]. Consistent with a previous study [17] investigating an MRI of the tissue features of the rabbit prostate complex in T1 and T2-weighted sequences, the present MRI anatomical study described the bulbourethral glands of the rabbit by creating a detailed image of the intensity of the studied structures in the mentioned sequences.

The MRI of bulbourethral glands in rabbits has been characterized by relative intensity and homogeneity, in contrast to the findings for these glands in the man. The detection of imaging glandular findings has been evidence to eliminate ambiguities regarding the localization of soft tissue findings in the perineal region arising from prostate hypertrophy, prostate cancer or Syringocele, Cobb’s collar, cowperitis, or lithiasis [19,20,21,25]. Additionally, the rabbit has been evaluated as a suitable animal model for the imaging study of urethral and paraurethral lesions, similar to the mouse and other animal species [10,18,22,23,24,26,31]. The current study has confirmed the information of different research [28,29] regarding the increasing importance of MRI as a method for imaging different parts of the urogenital system. The conducted MRI study of the rabbit bulbourethral glands was of an anatomical type and similar to previous investigations [33], providing anatomical images for the determination of the intactness or pathological origin of imaging findings in the pelvic cavity. Therefore, this imaging study has presented an anatomical part of the methodological algorithm that is required for the transition from an imaging norm to imaging pathology [33,34,35,36].

The given investigation has been related to all those aforementioned, which support the notion that MRI could improve knowledge for the assessment of the status of the reproductive system in other farm animals [23].

The final analysis indicated that the transverse (axial) T2-weighted imaging of the bulbourethral glands in the rabbit achieves detailed images that are of higher anatomical contrast compared to images obtained in T1-weighted sequences. The hyperintensity of the glandular findings at the T2-weighted sequence, relative to the adjacent soft tissues, was due to the content of secretory fluids in their glandular parenchyma. On dorsal imaging, the bulbourethral glands were identified as perineal, soft tissue findings with defined organ localization, topography, and intensity that clearly differentiate them from peripheral organs at the pelvic outlet.

The sagittal MRI of the rabbit bulbourethral glands is dependent on the localization of the research plane relative to the median plane. The decrease in the distance between the sagittal and the median plane caused a decrease in the imaging glandular detail. This was due to the localization of these glands as paired organs on a dorsolateral aspect to the end of the pelvic urethra. In the study, organ boundaries were successfully marked on T2-weighted images, in contrast to the lack of distinct MRI features characteristic of the glandular stroma and parenchyma. The sagittal MRI achieved the precise identification of the glandular topography and localization, as the anatomical contrast was enhanced on T2-weighted sequences.

The MRI of rabbit bulbourethral glands has allowed for a selection of solid and soft tissue anatomical landmarks in order to differentiate them from adjacent soft tissues in the retroperitoneal part of the pelvic cavity. The obtained imaging results could be used as a morphological base for clinical practice and reproduction.

## 5. Conclusions

MRI is a definitive and innovative method with which to study bulbourethral glands. These imaging results would be used as an anatomical base for clinical practice and reproduction. The anatomical approach of the present findings contributes to the imaging differentiation of the glands from neighboring soft tissue structures in the retroperitoneal part of the pelvic cavity.

## Figures and Tables

**Figure 1 animals-13-01519-f001:**
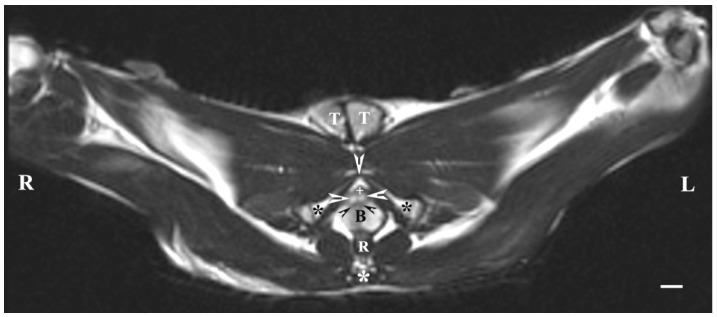
T2-weighted transverse image of rabbit pelvis in the plane through the level of the first caudal vertebra (white star): R—right; L—left. Bulbourethral glands (white horizontal arrows), the pelvic part of the urethra (white cross), penile bulb (B), rectum (R), ischial bones (black stars), the caudal part of the pelvic symphysis (white perpendicular arrow), pelvic diaphragm (black inclined arrows), and testes (T). Line—10 mm.

**Figure 2 animals-13-01519-f002:**
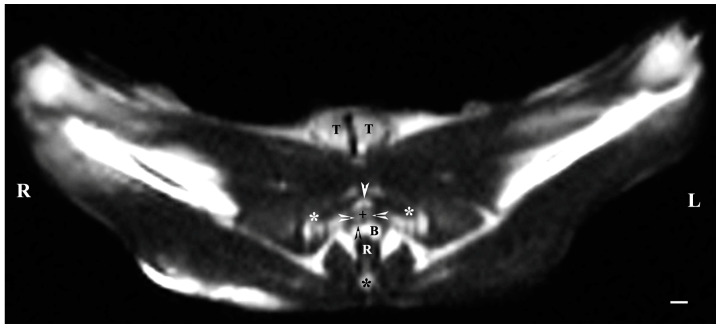
T1-weighted transverse image of rabbit pelvis at the level of the first caudal vertebra (black star): R—right; L—left. Bulbourethral glands (white horizontal arrows), the pelvic part of the urethra (black cross), penile bulb and penile root (B), rectum (R), ischial bones (white stars), the caudal part of the pelvic symphysis (white perpendicular arrow), pelvic diaphragm (black perpendicular arrow), and testes (T). Line—10 mm.

**Figure 3 animals-13-01519-f003:**
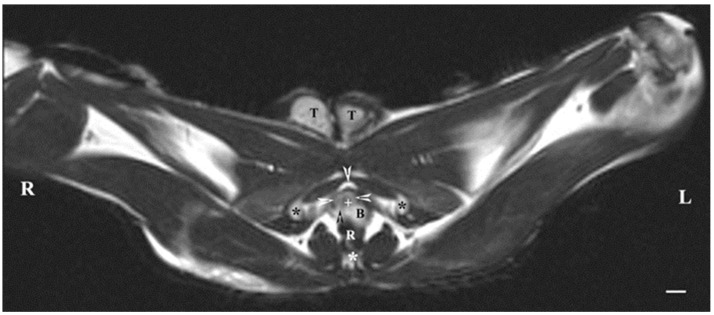
T1-weighted transverse image of rabbit pelvis through the level of the second caudal vertebra (white star): R—right; L—left. Bulbourethral glands (white horizontal arrows), the pelvic part of the urethra (white cross), penile bulb (B), rectum (R), ischial bones (black stars), the caudal part of the pelvic symphysis (white perpendicular arrow), pelvic diaphragm (black perpendicular arrow), and testes (T). Line—10 mm.

**Figure 4 animals-13-01519-f004:**
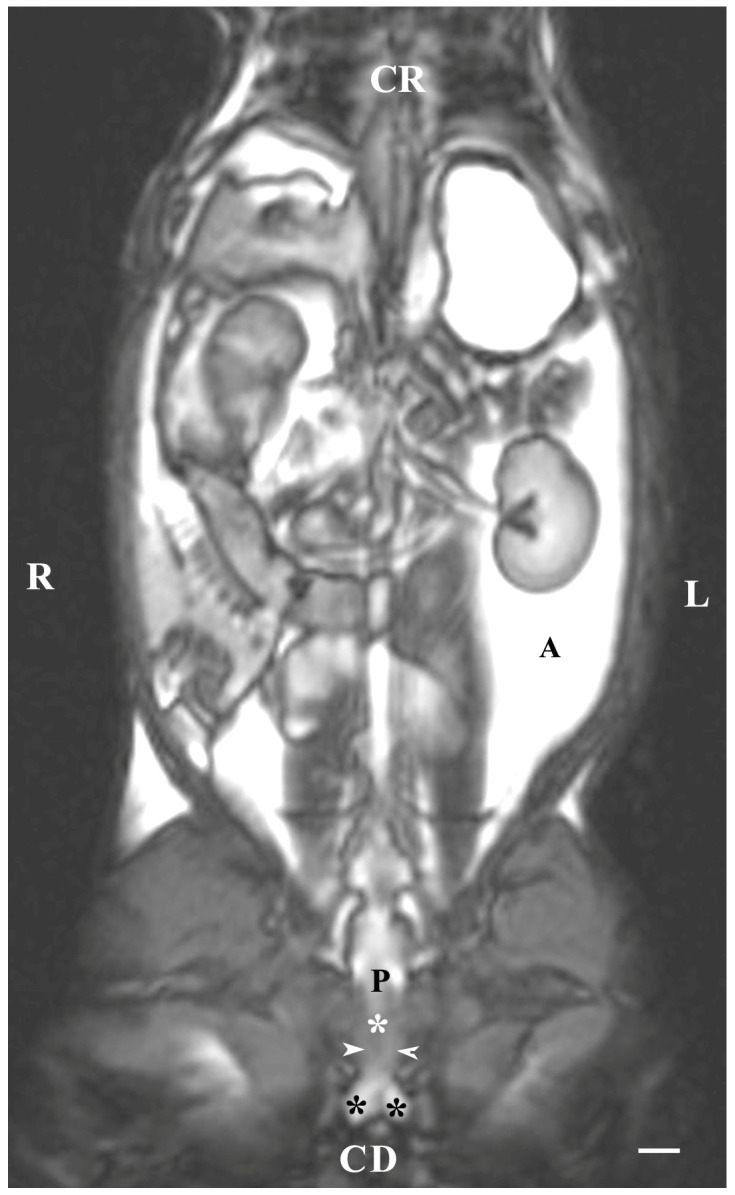
T1-weighted dorsal image of rabbit pelvis (at the level to the greatest distance, dorsal to the pelvic symphysis): R—right; L—left; CR—cranial; CD—caudal. Bulbourethral glands (white horizontal arrows), prostate gland complex (P), the pelvic part of the urethra (white star), root of the penis (black stars), and adipose tissue (A). Line—10 mm.

**Figure 5 animals-13-01519-f005:**
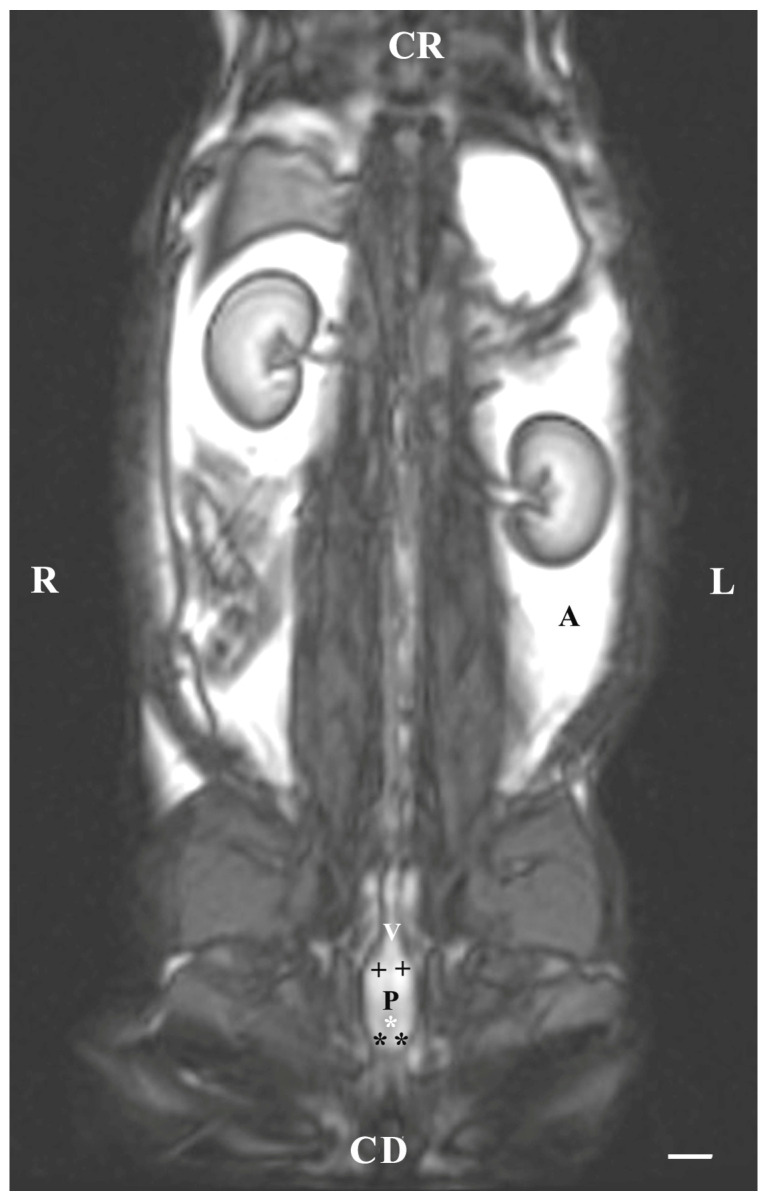
T1-weighted dorsal image of rabbit pelvis (at the level of the middle distance, dorsal to the pelvic symphysis): R—right; L—left. CR—cranial; CD—caudal. Bulbourethral glands (black stars), prostate complex (P), vesicular glands (black cross), the pelvic part of the urethra (white star), neck of the urinary bladder (V), and adipose tissue (A). Line—10 mm.

**Figure 6 animals-13-01519-f006:**
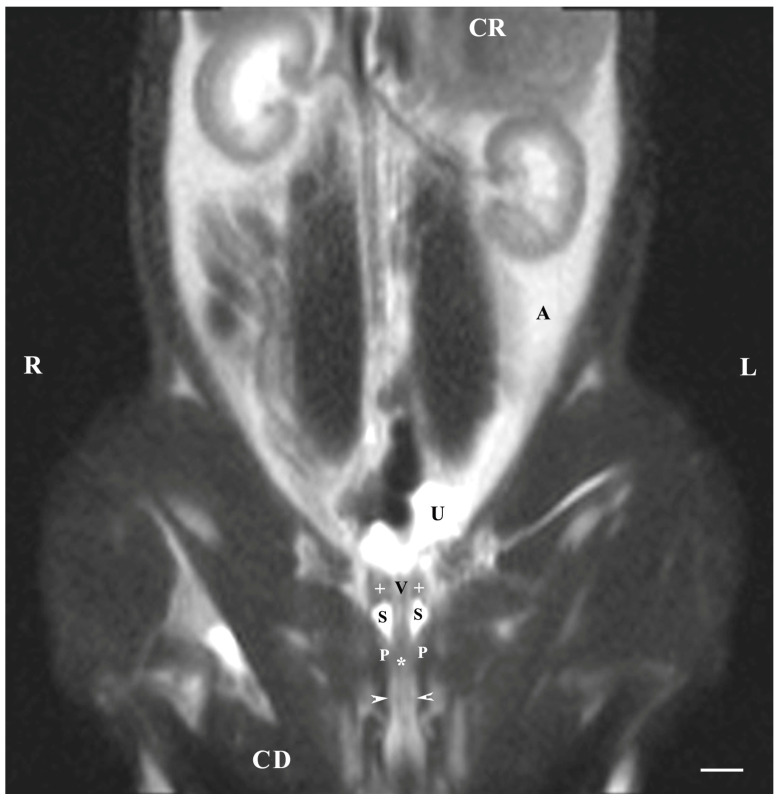
T2-weighted dorsal image of rabbit pelvis (at the level, close to the pelvic symphysis): R—right; L—left; CR-cranial; CD—caudal. Bulbourethral glands (horizontal white arrows), prostate complex (P), vesicular glands (white cross), pelvic symphysis (S), neck of the urinary bladder (V), the pelvic part of the urethra (white star), urinary bladder (U), adipose tissue (A). Line—10 mm.

**Figure 7 animals-13-01519-f007:**
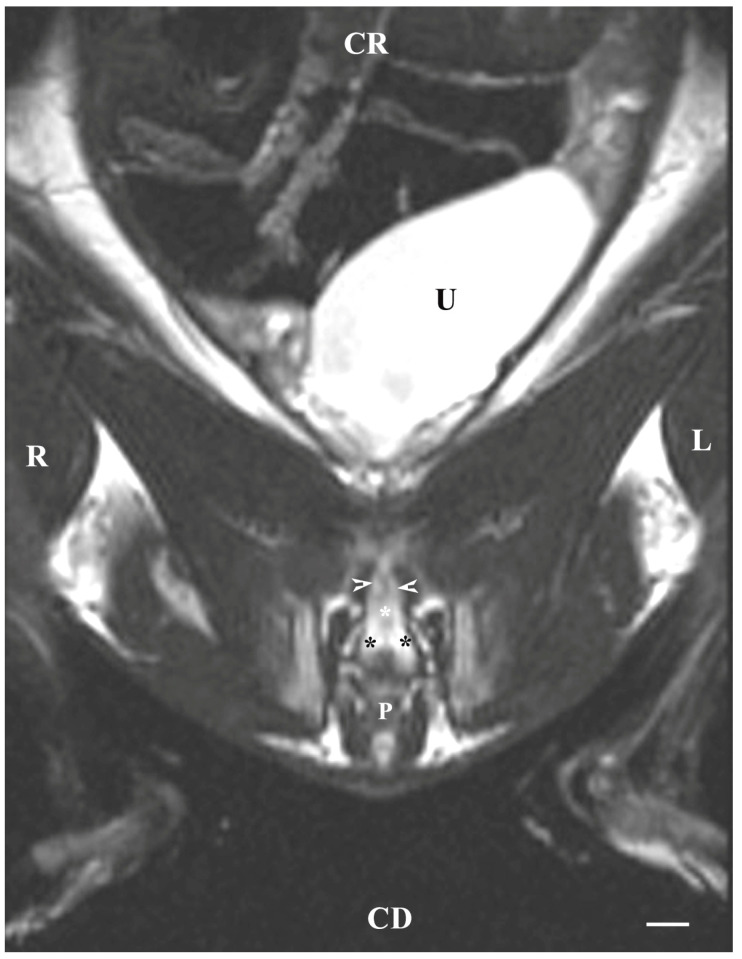
T2-weighted dorsal image of rabbit pelvis (at a greater distance, at the level to the pelvic symphysis): R—right; L—left; CR—cranial; CD—caudal. Bulbourethral glands (black stars), caudal part of the prostate gland complex (white horizontal arrows), the pelvic part of the urethra (white star), urinary bladder (U), and penis (P). Line—10 mm.

**Figure 8 animals-13-01519-f008:**
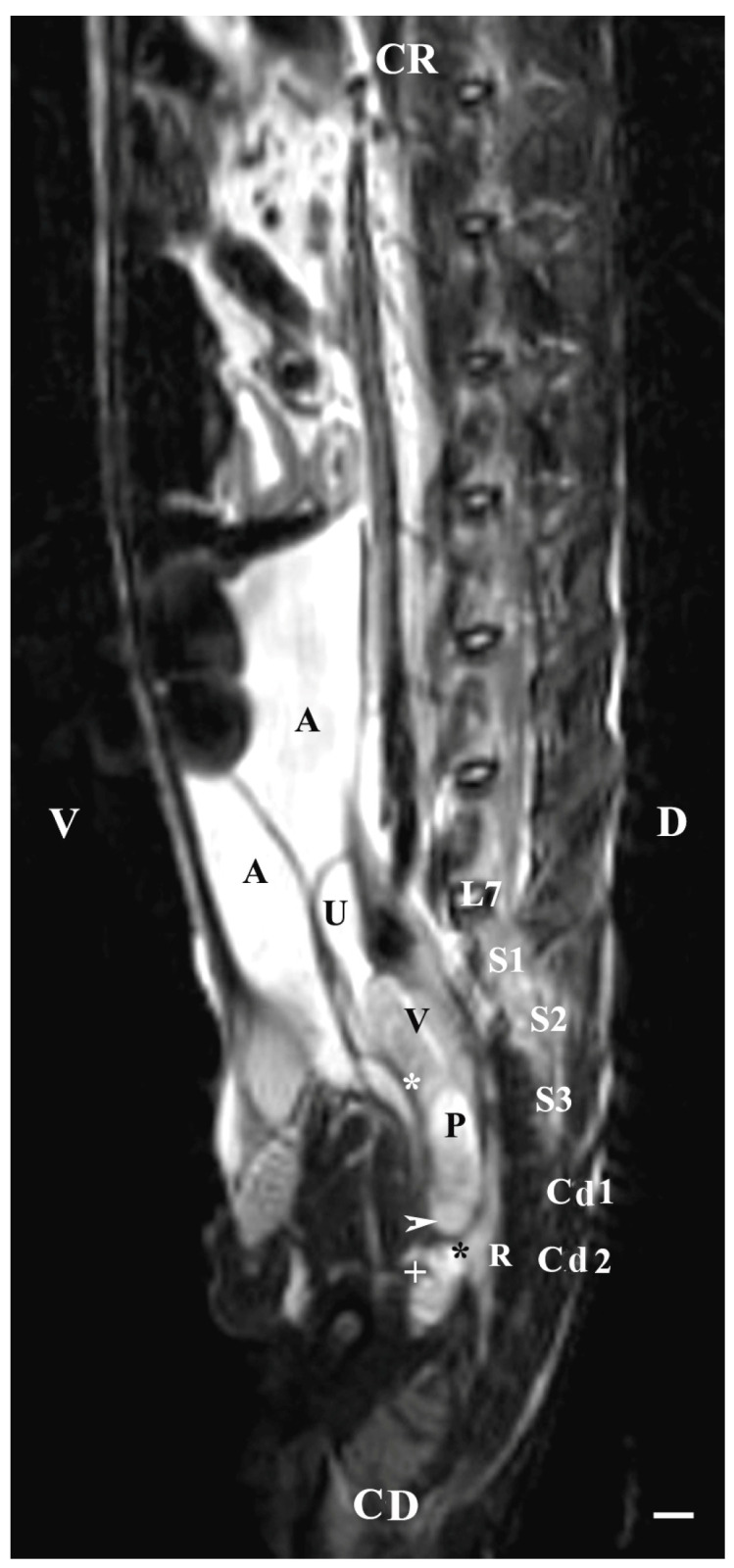
T2-weighted sagittal image of rabbit pelvis: V—ventral; D—dorsal; CR—cranial; CD—caudal. Bulbourethral glands (horizontal white arrow), vesicular glands (V), prostate gland complex (P), the pelvic part of the urethra (white star), the caudal part of the urinary bladder (U), adipose tissue (A), rectum (R), pelvic diaphragm (black star), root of the penis (white cross), L7—seventh lumbar vertebra, S1—first sacral vertebra, S2—second sacral vertebra, S3—third sacral vertebra, Cd1—first caudal vertebra, and Cd2 second caudal vertebra. Line—10 mm.

**Figure 9 animals-13-01519-f009:**
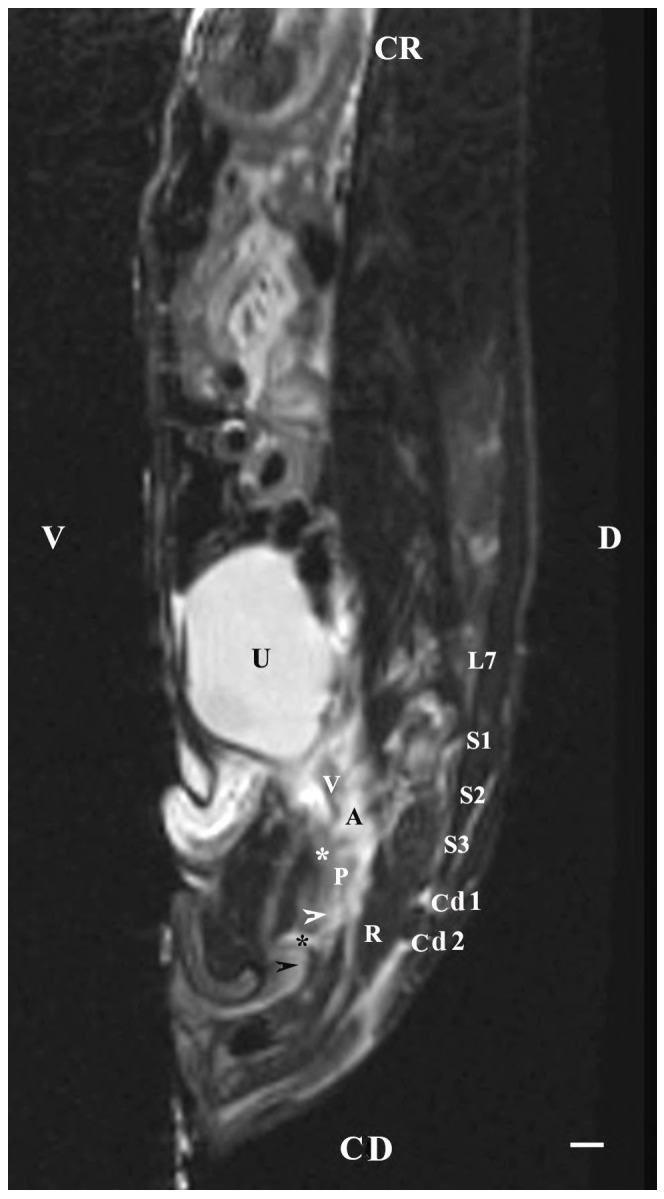
T1-weighted sagittal image of rabbit pelvis: V—ventral; D—dorsal; CR—cranial; CD—caudal. Bulbourethral glands (horizontal white arrow), the pelvic part of the urethra (white star), vesicular glands (V), prostate complex (P), pelvic diaphragm (black star), root of the penis (black horizontal arrow), urinary bladder (U), adipose tissue (A), rectum (R), L7—seventh lumbar vertebra, S1—first sacral vertebra, S2—second sacral vertebra, S3—third sacral vertebra, Cd1—first caudal vertebra, and Cd2 second caudal vertebra. Line—10 mm.

## Data Availability

All data are included in the present manuscript.

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
