# Peer review of "MRI Anatomical Investigation of Rabbit Bulbourethral Glands"

_animals, 2023, doi:10.3390/ani13091519_

Round 1

Reviewer 1 Report (Previous Reviewer 2)

Thank you for the corrections you made to the first manuscript version. However, minor changes are still required.  The names of the imaging planes were changed in Materials and Methods to transversal, dorsal and sagittal. However, the term "median" is often used later in the text and is incorrect (median = sagittal plane through the body's centerline). Please, change the word median to sagittal (= all planes parallel to the body's centerline).  Line 40: Please, remove the words "signal-intense". Line 41: Please, remove the word "signal". Line 50: I ask you to use keywords other than words already used in the title.  Lines 205-219: Thank you for adding some necessary information to this paragraph, but it is still imperfect and has some contradictory information. The following sentences are unnecessary and could be removed: "Active Shielding; homogeneity of field of view; the resonance frequency of a hydrogen proton in 1.5 T was 64 MHz". On line 209, you provide the pixel dimension (1 mm2) and the voxel size as (10 mm3) at the end. These two values don't fit together. Am I missing something? The TR value is written twice in the paragraph (lines 215 and 217). I don't understand the following sentence (lines 212-213): "2D acquisition schemes were applied for the sequences; FOV was: 50 x 50 x 50 cm (with mean values 250/250), in all directions". I understand the information about FOV, but not the rest. 

Author Response

Rosen Dimitrov

Professor, PhD

[email protected]

Kamelia Stamatova-Yovcheva

Associate Professor, PhD

[email protected]

Animals/MDPI

Editor

05.04.2023

Dear reviewer,

First, we highly appreciate your positive evaluation of the scientific content of our work. Also, we thank you for the useful comments. They gave us some positive criticism for reviewing and improving our manuscript.

Considering all your notices, we have made recommended changes that are included in the revised manuscript in Track changes mode.

  1. Firstly, we changed the terms of the mentioned planes. The term “median” was replaced correctly by the term “sagittal”. You mentioned correctly that sagittal is used when the planes are parallel to the centerline.
  2. Considering with your remarks about the interpretations “signal-intense” and “signal”, they were removed.
  3. The keywords were changed, according to your opinion.
  4. The following sentences “Active Shielding; homogeneity of field of view, the resonance frequency of a hydrogen proton in 1.5 T was 64 MHz” was removed.
  5. Regarding the pixel dimensions and the voxel size at the end of the section “Materials and Methods, we have added the following explanation: “The size of the voxel was increased to point the amount of tissue components of the glands. In the same time the number of the nuclei was increased parallel to the SNR [25. Kastler, B.; Vetter, D.; Patay, Z.; Germain, P. Chapter 5. Contrast in T1, in T2 and proton density. In: Physical Principles of Magnetic Resonance Imaging. Translated from the fifth edition and edited by V. Hadjidekov, Medicine and Physical Ed-ucation, Sofia, 2005, pp. 35-58].
  6. The term TR was doubled and we corrected this repetition.

As whole the section “Materials and Methods” was strictly revised. We were consulted by the technical staff of MRI laboratory.

Best regards:

The authors

Reviewer 2 Report (Previous Reviewer 4)

The authors have made a great job in the revised copy which in my opinion can be accepted 

Author Response

Rosen Dimitrov

Professor, PhD

[email protected]

Kamelia Stamatova-Yovcheva

Associate Professor, PhD

[email protected]

Animals/MDPI

Editor

05.04.2023

Dear reviewer,

First, we highly appreciate your positive evaluation of the scientific content of our work. Also, we thank you for your scientific attitude, regarding the structure of the manuscript, the used terms and literature data.

Best wishes:

The authors

Reviewer 3 Report (New Reviewer)

It is an interesting idea to present information on the bulbourethral glands in the rabbit. There is little information about the bulbourethral glands of a rabbit in the literature.However, the text contains some shortcomings:

The text is, however, hard to read and needs both English and overall design correction.

The images are low quality and should be cropped to the described area.

The study would be more interesting with other anatomical methods (classical dissection) added.

Images and descriptions comparing the results obtained with different techniques should be included.

The study group should be more numerous. Besides, the beginning of the paragraph mentions 12 rabbits and the end mentions 5. What does the information about the 12 rabbits refer to?

It is worth confirming that there were no pathological changes in the examined glands, for example, by histopathological examination. The fact that clinically the rabbit is healthy does not mean that there can not be changes in the examined glands.

The discussion section could involve more information on accessory glands of different species.

Keywords should not be a repetition of the words in the title.

Author Response

Rosen Dimitrov

Professor, PhD

[email protected]

Kamelia Stamatova-Yovcheva

Associate Professor, PhD

[email protected]

Animals/MDPI

Editor

05.04.2023

Dear reviewer,

First, we highly appreciate your positive evaluation of the scientific content of our work. Also, we thank you for the useful comments. They gave us some positive criticism for reviewing and improving our manuscript.

Considering all your notices, we have made recommended changes that are included in the revised manuscript in Track changes mode.

  1. We made major English language revisions, consulting native English speaker specialist from Trakia University, Stara Zagora, Bulgaria.
  2. The images have been obtained from the described MRI device. They were given in this format, for better precision of the topography of the studied glands as well the used anatomical landmarks for the glands’ localization. The image of the trunk has been cropped in a level that allows better anatomical space orientation to point the important anatomical landmarks in the caudal part of the trunk. In the sagittal and dorsal plane this is of great importance to demonstrate the trunk as whole.
  3. Regarding your remarks “The study would be more interesting with other anatomical methods (classical dissection) added”.

We describe the MRI imaging anatomical specifics of the rabbit bulbourethral glands, using only live healthy animals. Thus we prove the use of the imaging modalities as self-life, protective techniques, by which the life of the studied objects is protected. This algorithm is considered with the rules of the European legislation for animal protection and welfare.

  1. Conventional Anatomy of the rabbit is described in details by Barone [7. Barone R., Splanchnologie II. In: Anatomie comparée des mammifères domestiques. 3E edition; Baronne, R. Еd.; Vigot, Paris, 2001; Tome 4, pp. 159-185]. Thus, we used the experience of the authors as anatomical base to study the live imaging anatomical specifics of the glands. The author is cited in the text. Our aim is to investigate the morphological specifics of the rabbit bulbourethral glands in live animals, using non-invasive, protective method.
  2. The used animals are twelve, clinically healthy, male live rabbits, aged eight months (New Zealand white breed), with weight from 2.8 kg to 3.2 kg. Please, look line 169-170. We never mentioned five animals.
  3. Regarding the involving of histopathological results, our manuscript is focused only on macroscopic, anatomical imaging features of the studied glands, no microscopic histopathologic specifics.
  4. We involved in Discussion section only the specific of the glands in farm animals and man, because the rabbit is used often as anatomical models in Human Medicine.
  5. The keywords are corrected in accordance to your recommendations. They do not repeat the words in the title of the manuscript.

Best regards:

The Authors

Round 2

Reviewer 3 Report (New Reviewer)

The manuscript looks better. I believe the value of the paper would be much greater if MRI and autopsy images of the same individuals were shown. However, despite the use of only one method, this paper is of great value to clinicians.

Author Response

Response to the reviewer

Rosen Dimitrov

Professor, PhD

[email protected]

Kamelia Stamatova-Yovcheva

Associate Professor, PhD

[email protected]

Animals/MDPI

Editor

20.04.2023

Dear reviewer,

First, we highly appreciate your positive evaluation of the scientific content of our work. Also, we thank you for the useful comments. They gave us some positive criticism for reviewing and improving our manuscript.

Considering all your notices, we have made recommended changes that are included in the revised manuscript in Track changes mode.

  1. Regarding your remarks about the use of histological methods and images, we explain that such results are involved as “Golden standard” when ultrasound images are interpreted. We have such published investigations in an article Histological definition for the gray scale ultrasonography of the rabbit liver with authors Kamelia Stamatova-Yovcheva, Rosen Dimitrov, David Yovchev, Diyana Vladova, Omer Gurkan Dilek, Radoslav Mihaylov, 2018. It was published in Vet Hekim Der Derg 89 (1): 32-41. A histologist was included.

In this manuscript we only interpret the anatomical topographic features of the rabbit bulbourethral glands, using Magnetic Resonace Imaging. Our aim has been directed on the imaging anatomical and macroscopical characteristics of the studied glands.

  1. Regarding your notes about the number of the animals for the images, we have collected and presented the images, obtained from all twelve animals. The obtained results were exported and imported by DICOM and USB devices. They were adequately selected in accordance to the studied anatomical planes and landmarks.

This manuscript is a resubmission of an earlier submission. The following is a list of the peer review reports and author responses from that submission.

Round 1

Reviewer 1 Report

Review: paper by Dimitrov R and Stamatova-Yovcheva

The paper describes anatomical investigations of the rabbit bulbourethral glands. The study was conducted on healthy male rabbits, and the anatomical images were based on MRI data. A standard clinical MRI system was used. Different MRI sequences (T1-, T2-weighted sequences) were employed and the reconstructed 2D images were viewed and discussed. Examples of the acquired images are shown and described.

I have some general problems with this paper:

-        There is a general lack of aim(s). I do not find a specific aim anywhere in the Introduction section, neither in abstract section. So, I speculated whether the aim is to use the rabbits as a model to optimize MRI protocols for imaging of the bulbourethral glands. If yes, then the authors - in a more convincing way – should describe why the specific MRI protocols were used. On the other hand, if the aim is to describe rabbit bulbourethral glands from a zoological/veterinarian point of view, such investigations such be compared to findings revealed golden standard methods (histology).

-        Why were n=8 used in the study when no quantitative measures were included? The relatively large number of animals could be used to describe the anatomical variations, for example length, volume, etc, for the rabbit bulbourethral glands. Unfortunately, the authors did not include quantitative measures.

-        The MRI lacks a co-author with MRI experience. I urge the authors to reach out to an MRI expert to rewrite the MRI section. For example, there is no information about the used MRI RF-coil (which is important to understand the signal-to-noise ratios), the details about the T1- and T2- weightings are difficult to understand. My questions are (for example): were the MRI sequences performed using 2d or 3d acquisition schemes, what are the acquired image (voxel) resolutions, etc…. And some sentences do not really not make sense, such as “1131 images per second at high resolution, simultaneously in dorsal….” (page 4). That cannot be true.

-        It looks as Fig 1 is more clear than Fig 2 and Fig 3. Why?

-        Why are the authors using the word “pre-clinical”? I suggest to omit that word.

-        Again: why were no histological examinations performed to confirm MRI findings?

It is not clear why there is a both a “Simple summary” and an “Abstract”. Is that a requirement from the journal?

Reviewer 2 Report

The introduction is too long. Please, make it shorter.  The aim of the study/hypothesis (at the end of the introduction) is missing.  Lines 164-165: "...caudal extension of the pelvic limbs and flexion of their knee joints..." I am confused about the position of the hind legs. Were the limbs extended or flexed? It needs to be clearly written. Please, rephrase.  Line 165: The word "cadaver" should be used for a dead body. These animals were anesthetised, according to the text in the previous paragraph. Therefore, please, remove the word cadaver and use the appropriate word, e.g. animals.  Line 171: The terms transverse, dorsal and sagittal are commonly used in veterinary diagnostic imaging, and no other words are necessary. Please remove the words in brackets (for this paragraph and within the entire manuscript).  Line 172: The commonly used abbreviation for caudal vertebra is Cd. The abbreviation "C" is widely used for cervical vertebra. Please, change the acronym in the entire paragraph.  Lines 182-183: Information provided on these lines (field strength, magnet type) should be merged with the paragraph on lines 157-158 describing the MRI machine. Lines 183-185: I recommend removing these sentences—no practical value for the reader and probably specific for this type of machine.  Lines 185-186: I am not familiar with this specification of the matrix. There should be no negative numbers. Please, correct the numbers. Further at this line, the authors describe "proton resonance frequencies". The provided numbers are too high. The resonance frequency of a hydrogen proton in a 1.5T magnetic field is 64 MHz. The authors need to specify which proton/protons they describe. However, I think this has no value for the reader.  Lines 188 and 189: The terms "minimising contrast in T2" and "contrast minimisation in T1" are unclear. These terms are not routinely used to describe T1 or T2 sequences and should be removed. Please, provide some explanation if I didn't get it well.  Lines 182-191: This paragraph describes the imaging protocol, but some information regarding the sequences used in the study is missing. Which coils were used? Did they use only spin echo or also gradient echo sequences? What was the FOV size? SNR value? Duration of each sequence? Please, see the author guidelines of the journal Veterinary Radiology & Ultrasound at the following link: https://onlinelibrary.wiley.com/page/journal/17408261/homepage/diagnostic_imaging.htm (I know it is a different journal, but it is the official journal of the American College of Veterinary Radiology). Figures: I recommend putting alongside T1 and T2 weighted images in each plane. It should be better for the reader to get the complete image-related information. The orientation of the figures should be changed according to the commonly used orientation in veterinary medicine. Transversal plane images should have dorsal on the top of the figure. Sagittal plane images should have cranial on the left side of the figure and dorsal on the top. Both of the planes should have the right side of the animal on the left side of the image. The dorsal plane images are oriented correctly.  Line 352: Caudal is commonly abbreviated ad Cd. Please, change accordingly.  The chapter Conclusion is too long, and I recommend making it shorter. However, some of the information provided in the Conclusion should be used in the Discussion.  Line 555: Reference 32 is a PhD thesis. This type of literature doesn't belong to peer-review literature and should be avoided in a scientific report. Please, discuss it with the Editor and check the author's guidelines.

Reviewer 3 Report

The paper submitted by the authors provides important information about the bulbourethral glands. Nonetheless, the paper is not well organized, and almost all the sections provide information that is unclear.    Specific comments:   The simple summary does not provide adequate information about the study. in case animals were euthanized, why did not use anatomical cross-section?   The introduction section  In my opinion, this section is too long and provides unnecessary information. Here, you explain the main anatomy of Human bulbourethral glands. It is important to highlight that you are working with animals. Therefore, it would be better to compare with domestic or exotic species. In addition, you do not include references related to these statements. Line 69, mole? Lines 85-86, this sentence doesn’t make sense here, Please delete it. Lines 87-95, you do not need to explain the CT of it, nor the thickness of the slices. Lines 129-139, this paragraph would be better in the discussion. Line 161, please add the country and the brand. Line 165, you did not provide a previous explanation about euthanasia, then? Lines 182-191, I miss some MRI parameters such as FOV. In the results section, I miss a figure showing the level of sections. Line 206, hypointense or hypo intensive? The labelling of figures is not so easy to follow, you should use letters instead. Figure 2 doesn’t show good resolution. Figure 5, please confirm if it is T1W since the fluid looks hyperintense. In addition, CC is not correct, it should be CD. Lines 240-241, please check this sentence. In my opinion, is quite difficult to distinguish the shape and prominence of the glands in that figure.  Lines 247-249, all the structures display similar intensity (attenuation). Figures 9, 10, 11. Remember, it is an animal, not a human. Therefore, the figure disposition is not correct. In veterinary medicine, the head should be seen to the left and the tail to the right. Figure 9, the bulbourethral glands are labelled with perpendicular arrows, not horizontally in a correctly orientated figure. Figure 11, this figure doesn’t provide adequate information and should be deleted. The discussion and conclusion sections are quite disorganized and therefore, should be revised. Line 395, what mammals?  

Reviewer 4 Report

Dimitrov and Stamatova-Yovcheva carried out an research using  12 clinically healthy, sexually mature, New Zealand White male rabbits, 8 months of age, weighing 2.8 kg to 3.2 kg, using tunnel MRI equipment. The authors reported that Transverse (axial) MRI in T2-28 weighted sequence obtained detailed, signal-intense images that were of higher anatomical contrast than those in T1-weighted sequences. The signal hyperintensity of the glandular findings at T2, relative to the adjacent soft tissues, is due to the content of secretory fluids. The quality of the anatomical tissue contrast did not show much dependence on the choice of the sequence (T1 or T2) in dorsal MRI. Sagittal MRI of the bulbourethral glands in the rabbit is dependent on the localization of the research plane toward the median plane.  The investigators concluded that MRI of the bulbourethral glands in the NZW rabbit males is a necessary element for the study of the imaging differentiation of these findings from the adjacent soft tissue structures and the pathological alterations in the retroperitoneal part of the pelvic cavity. The Ms discuss an important topic provide useful information about the MRI of the bulbourethral glands in the rabbit.  In my opinion the the Ms can be improved considering with these items:

1.     Including color photos instead of black and white color

2.     The conclusion of the study may be improved further “MRI is a definitive and innovative method to study the rabbit bulbourethral glands. The imaging results can be used as a morphological base for the clinical practice and the reproduction. The anatomical approach of the present findings can help for imaging differentiation from the adjacent soft tissue structures and pathological alterations in the retroperitoneal part of the pelvic cavity”.

3.      References for 2022 and 2023, can improve the output of this MS and updating information.

4.      The discussion may be improved further by connecting the present findings with those found in the other farm animals for application.